# Gross Chromosomal Rearrangement at Centromeres

**DOI:** 10.3390/biom14010028

**Published:** 2023-12-24

**Authors:** Ran Xu, Ziyi Pan, Takuro Nakagawa

**Affiliations:** 1Department of Biological Sciences, Graduate School of Science, Osaka University, 1-1 Machikaneyama, Toyonaka 560-0043, Osaka, Japan; 2Forefront Research Center, Graduate School of Science, Osaka University, 1-1 Machikaneyama, Toyonaka 560-0043, Osaka, Japan

**Keywords:** centromere, DNA repeat, heterochromatin, CENP-A, DNA recombination, DNA repair, gross chromosomal rearrangement, fission yeast, humans, isochromosome

## Abstract

Centromeres play essential roles in the faithful segregation of chromosomes. CENP-A, the centromere-specific histone H3 variant, and heterochromatin characterized by di- or tri-methylation of histone H3 9th lysine (H3K9) are the hallmarks of centromere chromatin. Contrary to the epigenetic marks, DNA sequences underlying the centromere region of chromosomes are not well conserved through evolution. However, centromeres consist of repetitive sequences in many eukaryotes, including animals, plants, and a subset of fungi, including fission yeast. Advances in long-read sequencing techniques have uncovered the complete sequence of human centromeres containing more than thousands of alpha satellite repeats and other types of repetitive sequences. Not only tandem but also inverted repeats are present at a centromere. DNA recombination between centromere repeats can result in gross chromosomal rearrangement (GCR), such as translocation and isochromosome formation. CENP-A chromatin and heterochromatin suppress the centromeric GCR. The key player of homologous recombination, Rad51, safeguards centromere integrity through conservative noncrossover recombination between centromere repeats. In contrast to Rad51-dependent recombination, Rad52-mediated single-strand annealing (SSA) and microhomology-mediated end-joining (MMEJ) lead to centromeric GCR. This review summarizes recent findings on the role of centromere and recombination proteins in maintaining centromere integrity and discusses how GCR occurs at centromeres.

## 1. Introduction

The centromere region of chromosomes is essential in eukaryotes to ensure faithful chromosome segregation. A histone H3 variant CENP-A (or cenH3) specifically localizes at the centromere [1,2]. The chromatin containing CENP-A, called the CENP-A chromatin, recruits the constitutive centromere-associated network (CCAN) proteins and, in turn, the KMN complex to form the large kinetochore protein complex [3,4]. While a single CENP-A nucleosome is sufficient to build the kinetochore on a point centromere in budding yeast, multiple CENP-A nucleosomes are involved in the kinetochore formation in the case of regional centromeres [5,6]. CENP-A is conserved through evolution and is present in animals, plants, and fungi [1,7,8]. However, the nucleotide sequence of the centromere is not conserved. Repetitive DNA sequences, including satellite repeats and transposable elements (TEs), are present in the centromere in a variety of eukaryotes [9,10,11,12]. Recent advances in DNA sequencing and assembly techniques have revealed the complete sequence of the human genome from telomere to telomere (T2T) [13,14,15]. However, the biological meaning of having repetitive elements at the centromere remains unclear. In addition to the CENP-A chromatin, heterochromatin characterized by di- and tri-methylation of H3K9 (H3K9me2,3) assembles on repetitive DNA sequences at the centromere [16]. These epigenetic marks are essential for the correct segregation of chromosomes in mitosis and meiosis [17,18]. Moreover, recent studies uncovered their role in suppressing gross chromosomal rearrangement (GCR) at the centromere. Robertsonian translocation between acrocentric chromosomes is the most frequently observed chromosomal abnormality in humans. Isochromosomes, whose arms mirror each other, are produced by recombination between inverted repeats present at a centromere and commonly observed in cancer cells [19]. Isochromosomes may accelerate tumor growth by altering gene dosage. It is also possible that a centromere becomes unstable in cancer cells [20,21]. This review summarizes the centromere sequence and chromatin in humans and the fission yeast *Schizosaccharomyces pombe* and discusses recent findings on how centromere integrity is maintained and how GCR occurs at the centromere.

## 2. Chromatin Structures and DNA Repeats at Centromeres

### 2.1. The Human Centromere

Complete genomic DNA and epigenetic maps of human centromeres have been reported in 2022 [22,23] (Figure 1). Human centromeres contain tandem arrays of alpha-satellite (αSat) repeat of ~171 bp. αSat repeats sometimes form higher-order repeats (HORs). Fifteen autosomes and the two sex chromosomes have unique centromeric αSat HOR arrays, and the rest can be grouped into two families based on similarity (chr1, 5, and 19; chr13, 14, 21, and 22). While 18 of the 23 chromosomes contain multiple arrays of αSat HORs, only one αSat HOR per chromosome is active and associates with CENP-A. Holliday junction-recognizing protein (HJURP) and Mis18 deposit CENP-A to the centromere in the G1 phase of the cell cycle [24,25,26,27]. CENP-A nucleosome interacts with the CCAN complex of 16 subunits, including CENP-C and CENP-T [28,29,30]. The CCAN complex, in turn, recruits the KMN complex containing Knl12, Mis12, and Ndc80, which captures the end of spindle microtubules. Active HORs range from 340 kb (chr21) to 4.8 Mb (chr18) in length. Non-coding RNAs transcribed from the core region of human centromeres act in cis to contribute to the centromere function [31,32].

Heterochromatin characterized by H3K9me2,3 is formed on pericentromeric repeats [33] (Figure 1). The Suv39h lysine methyltransferase plays a major role in the H3K9 methylation at the centromere [34]. The chromodomain protein that recognizes H3K9me2,3 marks, including the heterochromatin protein 1 (HP1), inhibits RNA polymerase II (RNAPII)-dependent transcription [35,36]. Another epigenetic mark of heterochromatin, CpG DNA methylation, is abundant in the human centromere [37]. αSat HORs are often surrounded by αSat monomers, other types of satellite repeats, such as beta-satellite (βSat), hSat1, hSat2, and hSat3, transposable elements, and segmental duplications (Figure 1). However, the function of the pericentromeric repeats is poorly understood. Distinct repetitive variants may arise within each centromere and expand through successive tandem duplication. Centromeric and pericentromeric repeats together account for 6.2% of the genome (190 Mb) [22] (Figure 2), which is four times longer than protein-coding sequences [13]. Centromere repeats are not always arranged in tandem but are placed in inversion. A 1.7 Mb inversion is present in the active αSat HOR array on chr1 [22]. Inversions also exist in the inactive αSat HOR on chr3, chr16, and chr20. More than 200 inversions exist in hSat3 arrays on chr9 and βSat arrays on chr1 and acrocentric chromosomes. DNA recombination between centromere repeats can result in GCR, including translocation, inversion, and isochromosome formation. Robertsonian translocation occurs between two acrocentric chromosomes and is the most frequently observed chromosomal abnormality in humans (1/1,000 individuals) [38]. The short arms of the human acrocentric chromosomes, chr13, 14, 15, 21, and 22, contain ribosomal DNA (rDNA) repeats and pseudo-homologous regions [39]. Colocalizing rDNA repeats from different acrocentric chromosomes in a nucleolus can facilitate recombination between the adjacent pseudo-homologous regions. Recombination between the pseudo-homologous regions can occasionally result in Robertsonian translocation. Indeed, the pseudo-homologous regions contain the recurrent breakpoint of Robertsonian translocation. Isochromosomes whose arms mirror each other are produced by recombination at or around centromeres on the same chromosomes.

### 2.2. The Fission Yeast Centromere

Studies using the budding yeast *Saccharomyces cerevisiae* isolated the functional centromere sequence in eukaryotes for the first time [40]. Budding yeast has a point centromere with a unique sequence of ~120 bp occupied by a single nucleosome containing CENP-A [6,41]. On the other hand, many animals, plants, and a subset of fungi, including the fission yeast *Schizosaccharomyces pombe* and the pathogenic fungus *Candida albicans*, have regional centromeres occupied by multiple CENP-A nucleosomes. Regional centromeres sometimes contain repetitive DNA sequences. The fission yeast centromere contains DNA repeats, making it a useful model organism to study the nature of repetitive regional centromeres. The chromatin structure and DNA sequence of a fission yeast centromere are illustrated in Figure 3. In the fission yeast centromere, the central unique sequence (cnt) is surrounded by different types of centromere repeats, the inner-most repeat (imr), outer-repeat (otr) consists of dg and dh repeats, and the outer-most repeat (irc). The CENP-A chromatin assembles on the cnt and inner portions of the imr. 

Like humans, deposition of Cnp1, the CENP-A homolog, to the centromere chromatin occurs depending on Scm3, the HJURP homolog, and Mis18, although during the G2 phase of the cell cycle [42]. Active promoters and numerous transcriptional start sites exist in the CENP-A chromatin domain [43]. However, RNA transcripts are limited compared to RNA polymerase II (RNAPII) binding levels. Loss of Tfs1 or Ubp3 required to restart transcription increases CENP-A binding levels at the centromere, suggesting that RNAPII stalling initiates chromatin remodeling events that facilitate CENP-A deposition.

**Figure 3 biomolecules-14-00028-f003:**
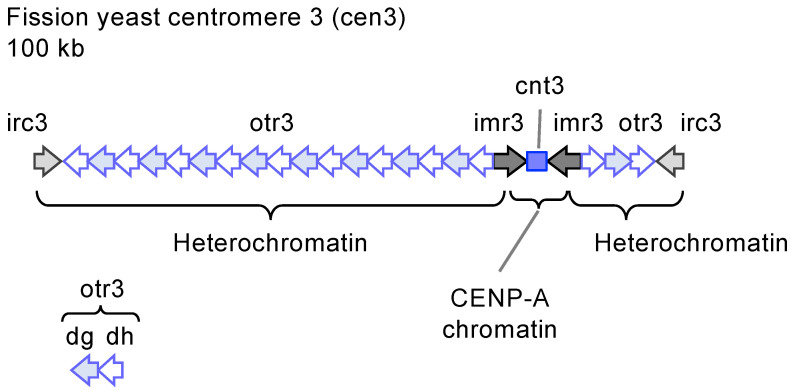
A schematic overview of chromatin structures and DNA sequences of the fission yeast centromere 3 (cen3). The copy number of otr3 containing dg and dh repeats can differ between laboratory strains [44].

Heterochromatin characterized by H3K9me2,3 assembles on imr outer portions, otr, and irc repeats. tRNA genes are present at the boundary of heterochromatin. The Clr4 lysine methyltransferase, the Suv39 homolog, is the sole enzyme required for H3K9me2,3 in fission yeast [16]. Centromere heterochromatin assembles in an RNA-interference (RNAi)-dependent manner. RNA-dependent RNA polymerase (Rdp1) forms double-stranded RNA using single-stranded RNA transcribed from centromere repeats as a template, and the Dicer ribonucleotide endonuclease (Dcr1) cleaves the double-stranded RNA, producing small RNA (sRNA) of ~22 bp [45,46]. The RNA-induced transcriptional silencing (RITS) complex containing Argonaute (Ago1) capturing sRNA localizes to the centromere and recruits the Clr4 methyltransferase [47,48,49]. Repetitive sequences promote RNAi-mediated heterochromatin formation [50]. In chicken cells, the centromeres containing repetitive sequences form heterochromatin, whereas those without repetitive sequences do not form heterochromatin [51,52], providing another link between DNA repeats and heterochromatin assembly. In addition to the RNAi mechanism, RNAPII pausing at centromere repeats promotes heterochromatin assembly [53,54]. No CpG DNA methylation has been reported in fission yeast.

## 3. Centromere Chromatin Maintains Centromere Integrity

Numerous factors related to centromere chromatin affect DNA transactions, including DNA double-strand break (DSB) repair (Table 1). We discuss the role of CENP-A chromatin and heterochromatin in maintaining centromere integrity.

**Table 1 biomolecules-14-00028-t001:** Centromere proteins involved in DNA repair and recombination.

Mammals	Fission Yeast	Centromere	DNA Transaction
CENP-A	Cnp1	Centromere-specific H3 variant	DSB localization, HR ^1^ [55,56,57]
CENP-N	Mis15	A CCAN component	DSB localization [55]
CENP-T	Cnp20	A CCAN component	DSB localization [55,56]
CENP-U	Mis17	A CCAN component	DSB localization [55]
CENP-I	Mis6	A CCAN component	HR [58]
CENP-S	Mhf1	A CCAN component	HR and GCR suppression [59,60]
CENP-X	Mhf2	A CCAN component	HR and GCR suppression [59,60]
CENP-B	Abp1/Cbh1/Cbh2	Centromere sequence binding	Replication fork stability [61]
HJURP	Scm3	CENP-A chaperone	HR [62]
MAD1	Mad1	Spindle assembly checkpoint	DDR ^2^ [63]
MAD2	Mad2	Spindle assembly checkpoint	DDR [63]
SMC1	Psm1	Cohesin	DDR [64]
SMC3	Psm3	Cohesin	DDR [64]
SMC2	Cut14	Condensin	HR [65]
SMC4	Cut3	Condensin	HR [65]
NCAPH	Cnd2	Condensin	DDR [66]
SMC5	Smc5	SMC protein	HR [67,68]
SMC6	Smc6	SMC protein	HR [67,68]
SUV39	Clr4	H3K9 methyltransferase	DDR, GCR suppression [69,70]
PHF2	Phf2	Histone demethylase	HR [71]
GCN5	Gcn5	Histone acetyltransferase	NER ^3^ [72]
PHF8	Epe1	H3K9 demethylase	DDR, GCR suppression [69,73]
UBE2A	Rhp6	H2B-K119 ubiquitin ligase	DDR [74]
KAT5	Esa1	H2A and H4 acetyltransferase	DDR [75]
AGO1, AGO2	Ago1	A RITS component	DDR, GCR suppression [69,76,77]
DICER1	Dcr1	Endoribonuclease Dicer	DDR, GCR suppression [69,78]
SIRT1	Sir2	Histone deacetylase	DDR, GCR suppression [69,79]
HDAC6/10	Clr3	Histone deacetylase	MMR ^4^, GCR suppression [69,80]
HDAC1/2	Clr6	Histone deacetylase	NHEJ ^5^, GCR suppression [69,81]
HP1	Swi6/Chp2	Chromodomain protein	HR, GCR suppression [69,82]
KAP1	Ngg1	Heterochromatin structure regulator	DDR [83]

^1^ HR, Homologous recombination. ^2^ DDR, DNA damage response. ^3^ NER, Nucleotide excision repair. ^4^ MMR, DNA Mismatch repair. ^5^ NHEJ, Non-homologous end joining.

### 3.1. The CENP-A Chromatin Maintains Centromere Integrity

The CENP-A chromatin maintains the centromere integrity in human cells. The chromosome-orientation fluorescent in situ hybridization (CO-FISH) showed that CENP-A depletion induces sister chromatid exchange at a centromere [56]. Lack of CENP-C, CENP-T, or CENP-W also induces centromere recombination, suggesting that CENP-A suppresses centromere recombination by recruiting the CCAN proteins to centromeres. On the other hand, Ndc80 depletion does not increase centromere recombination, although chromosome missegregation is elevated, suggesting that CENP-A and the CCAN proteins, but not the KMN complex, are specifically required to suppress centromere recombination.

CENP-A suppresses αSat transcription and DNA-RNA hybrid formation. CENP-A depletion in the S phase results in slow replication progression, accumulation of γH2AX indicative of DNA breaks, and hyper-recombination at centromeres [57]. Overexpression of a DNA-RNA hybrid-specific ribonuclease, RNaseH1, in CENP-A depleted cells reduces γH2AX and sister chromatid recombination at centromeres, suggesting that CENP-A maintains centromere integrity by suppressing DNA-RNA hybrid formation.

CENP-A and the CCAN proteins also play a role in DNA repair in non-centromeric regions of chromosomes. CENP-A, CENP-N, CENP-T, and CENP-U, are recruited to DSB sites and promote cell survival after DNA damage induced by ionizing radiation (IR) [55]. CENP-I, a member of the CENP-H/I/K complex, is also involved in DNA damage repair [58]. Loss of CENP-I reduces cell survival after exposure to IR. CENP-I depletion delays the disappearance of IR-induced 53BP1 foci, indicative of DNA damage, and impairs homologous recombination detected using the GFP reporter system in the U2OS cell line [84]. Loss of CENP-I also accumulates DNA-RNA hybrids, and RNaseH1 overexpression restored homologous recombination in CENP-I-depleted cells [58], suggesting that CENP-A and the CCAN proteins promote DNA recombination by suppressing DNA-RNA hybrid formation. Interestingly, exposure of murine cells to etoposide, a potent inducer of DSBs, specifically induced transcription of centromere repeats, and CENP-A is dissociated from the centromere [85]. The CENP-A dissociation from centromeres by DNA damage depends on p53 and DNA damage checkpoint kinase ATM. CENP-A might shuttle between centromeres and DNA damage sites.

### 3.2. Heterochromatin Maintains Centromere Integrity

Heterochromatin assembly depends on histone H3K9me2,3 and DNA methylation in mammals and plants. Knocking out the murine Suv39h lysine methyltransferase genes eliminates H3K9me2,3 at centromeres, resulting in chromosome instability and an increased risk of tumors [34]. Loss of DNMT3A and DNMT3B DNA methyltransferases in mice increases recombination between sister chromatids at centromeres [86], demonstrating the role of heterochromatin in suppressing recombination at centromeres. The Immunodeficiency, Centromere instability, and Facial anomalies (ICF) syndrome is characterized by DNA hypomethylation of pericentromeric satellite repeats and the formation of multiradial chromosomes at centromeres [87,88,89]. Recombination between centromere repeats on different chromosomes may form multiradial chromosomes. Mutation in DNMT3B, CDCA7, HELLS, ZBTB24, or UHFR1 results in DNA hypomethylation and causes ICF syndrome. CDCA7 and HELLS form a chromatin remodeling complex that interacts with UHFR1 to maintain the DNA methylation state [90,91,92]. ZBTB24 facilitates the expression of CDCA7 [93]. In those mutant cells, centromere transcription leading to DNA-RNA hybrid formation is induced, and the 53BP1 binding to αSat and hSat2 increases [90]. RNaseH1 overexpression reduces the 53BP binding to the centromere repeat, suggesting that heterochromatin maintains centromere integrity by suppressing DNA-RNA hybrid formation.

Heterochromatin characterized by H3K9me2,3 is formed in fission yeast, while no DNA methylation has been reported. Mutation in the Clr4 lysine methyltransferase or histone H3 at K9 increased isochromosome formation [69], suggesting that Clr4 suppresses isochromosome formation through H3K9 methylation. The chromodomain proteins that recognize H3K9me2,3: Swi6, Chp2, and Chp1, and the histone deacetylases: Sir2, Clr3, and Clr6 are involved in the GCR suppression, showing that heterochromatin suppresses GCR at centromeres. Loss of RNAi factors, including Ago1, increases isochromosome formation, demonstrating that the RNAi machinery plays a role in GCR suppression at centromeres. Interestingly, the mutation in the RNA polymerase II (RNAPII) catalytic subunit, Rpb1, or other transcription-related factors, including Mlo3, Tfs1/TFIIS, and Ubp3 reduces GCR in *clr4* deletion cells [69,94]. However, loss of transcription factors, including Ell1, Leo1, and Spt4, causes marginal effects on GCR in *clr4* deletion cells [69], showing that the function(s) specific to Mlo3, Tfs1, and Ubp3 is involved in isochromosome formation. Mlo3 RNA-binding protein facilitates the export of poly(A)+ RNA from the nucleus to the cytoplasm [95]. Yra1, the budding yeast homolog of Mlo3, binds DSB sites and facilitates DSB repair [96], suggesting that Mlo3 is directly involved in the GCR process. The progression of RNAPII can be paused and backtracked by DNA sequences and DNA-binding proteins. Following backtracking, Tfs1/TFIIS interacts with RNAPII and stimulates its RNA cleavage activity to re-initiate RNA synthesis [97]. Ubp3 ubiquitin protease also promotes RNAPII restart following backtracking by protecting RNAPII from ubiquitin-dependent degradation [98]. Therefore, Tfs1 and Ubp3 may cause centromeric GCR by restarting RNAPII following backtracking. Further analysis is required to address how these transcription-related factors cause isochromosome formation at centromeres.

Repair of CRISPR-Cas9-induced DSBs is temporally and spatially controlled in pericentric heterochromatin in mammals [99,100]. In the G1 phase, DSBs are positionally stable and recruit Ku80 involved in non-homologous end-joining (NHEJ). However, in the G2 phase, DSBs relocate to the periphery of heterochromatin, where Rad51 is recruited. DSB end resection by the MRN complex with CtIP is required for DSB relocalization toward the periphery. DSB relocalization from the heterochromatin domain to the nuclear periphery is also observed in *Drosophila* [101,102,103]. The repair of DSBs in the centromeric heterochromatin domain appears temporally and spatially regulated to prevent recombination between DNA repeats, resulting in deleterious GCR.

## 4. The Role of DNA Damage Checkpoint at Centromeres

### 4.1. DNA Damage Checkpoint Suppresses GCR at Centromeres

DNA damage checkpoint pathway regulates cell cycle progression and repair even in the absence of exogenous DNA damage [104]. In fission yeast, the Rad3/ATR checkpoint kinase is activated through the function of Replication Factor C (RFC)-like complexes containing Rad17 and the 9-1-1 complex containing Rad9, Rad1, and Hus1. Mutations in the Rad3 kinase increase isochromosome formation and translocation [105]. Mutations in Rad17 or the 9-1-1 complex increase the DSB-induced isochromosome formation [106]. DNA damage checkpoint response suppresses centromeric GCR.

### 4.2. The ATR Checkpoint Kinase Ensures Faithful Chromosome Segregation in Mitosis

In vertebrates, the ATR kinase localizes at centromeres on mitotic chromosomes through CENP-F to promote faithful chromosome segregation [107]. Aurora B kinase plays a role in correcting erroneous microtubule attachments in kinetochores [108,109]. ATR kinase is activated by Replication Protein A (RPA)-coated R-loops formed at centromeres, stimulating Aurora B kinase by activating the Chk1 kinase [107]. ATR is integrated into the regulation of microtubule attachments at the kinetochore.

## 5. Rad51-Dependent Homologous Recombination Safeguards Centromere Integrity

Rad51 is the key player in homologous recombination [110]. Single-stranded DNA is produced at stalled replication forks and at DSB ends. Rad51 binds single-stranded DNA to form nucleoprotein filaments and searches for homologous duplex DNA. Once Rad51 nucleoprotein filaments find a homologous duplex, Rad51 promotes DNA strand exchange, creating displacement-loop (D-loop). Homologous recombination mediated by Rad51 plays a vital role in maintaining genome integrity. Mutations in the factors promoting the Rad51-dependent recombination, such as BRCA1 and BRCA2, cause human breast and ovarian cancers [111]. Mutations in BRCAs and Rad51 are also found in Fanconi anemia patients characterized by physical abnormalities, bone marrow failure, and increased risk of malignancy [112]. Recombination between repetitive sequences can cause GCR. However, Rad51-dependent homologous recombination safeguards the integrity of the centromere containing DNA repeats.

### 5.1. Rad51 Suppresses Isochromosome Formation at Centromeres

Spontaneous DNA damage produced at a centromere can be repaired by homology-mediated DNA repair using centromere repeats as the template. Fission yeast produces isochromosomes using centromere inverted repeats (Figure 4: irc3, otr3, and imr3 inverted repeats) [105,113]. The pathogenic fungus *Candida albicans* also forms isochromosomes of chr5, resulting in hyper-resistant to azole, one of the antifungal drugs available [114]. Duplication of genes, including *ERG11* encoding the drug target, on the isochromosome 5 causes the azole hyper-resistant phenotype [115]. In humans, isochromosomes are commonly found in cancer cells [19].

In fission yeast, recombination between centromere inverted repeats can result in isochromosome formation (Figure 4, the right arrow). Detailed analysis of isochromosomes found their breakpoints present in centromere repeats [105,113], demonstrating that homology-mediated DNA recombination between centromere inverted repeats creates isochromosomes. At least one-third of isochromosomes formed in wild-type cells had their breakpoints in the heterochromatin domain [105], showing that recombination in heterochromatin can result in GCR. Rad51-dependent recombination safeguards centromere integrity (Figure 4, the left arrow), whereas Rad51-independent recombination causes GCR resulting in isochromosome formation (Figure 4, the right arrow) [117]. The loss of Rad51 greatly increased the spontaneous formation of isochromosomes [105]. Rad51 may process spontaneous DNA damage formed during DNA replication, as Rad51 binds to a centromere in the S phase [105]. Rad51 suppresses isochromosome formation even when a DSB is introduced outside a centromere [113]. Extensive degradation of DSB ends reaching a centromere repeat may initiate isochromosome formation. Rad51 paralogs Rad55 and Rad57 [118,119], the Mre11-Rad50-Nbs1 (MRN) complex [120], and Exo1 suppress the DSB-induced isochromosome formation. In addition to isochromosome formation, Rad51 suppresses chromosomal truncation with de novo telomere addition either inside or outside a centromere [121,122,123,124]. A growing number of DNA repair and recombination proteins have been shown to play roles at centromeres (Table 2).

### 5.2. Rad51 and Rad54 Promote Noncrossover Recombination at Centromeres, Thereby Suppressing Isochromosome Formation

Rad54, a SWI2/SNF2-type motor protein, is essential in homologous recombination in yeast [138,139,140]. Rad54 stabilizes Rad51 nucleoprotein filaments independently of ATP [141,142]. Rad54 facilitates Rad51-dependent homologous pairing and branch migration of joint molecules in an ATP-dependent manner [143,144,145]. Loss of Rad54 increased spontaneous isochromosome formation in fission yeast in a manner epistatic to *rad51* deletion [117], demonstrating that the Rad51-Rad54 axis of recombination suppresses isochromosome formation at the centromere.

How does the Rad51-Rad54-dependent recombination suppress isochromosome formation? The Rad51-Rad54-dependent recombination promotes noncrossover recombination, thereby suppressing isochromosome formation [117]. Homologous recombination can be divided into two modes: crossover and noncrossover recombination. The reciprocal or nonreciprocal exchange of chromosome regions defines crossover recombination. In contrast to crossover, the flanking regions do not exchange in noncrossover recombination. Physical analysis of recombination products revealed that Rad51 and Rad54 preferentially promote noncrossover recombination between DNA repeats at a centromere [117]. Crossover recombination between centromere inverted repeats likely results in isochromosome formation or the inversion of the central region (i.e., cnt). Indeed, the loss of Mus81, a structure-specific DNA endonuclease that preferentially causes crossover [146,147,148], reduces the isochromosome formation and inversion in *rad51* deletion cells [117]. 

It appears that noncrossover recombination is an intrinsic feature of the Rad51-Rad54-dependent recombination. An amino acid substitution of the evolutionally conserved lysine in the ATP-binding domain P-loop, *rad54-K300A,* does not abolish yeast two-hybrid interaction between Rad51 and Rad54. The *rad54-K300A* mutation accumulates spontaneous Rad54 foci in a Rad51-dependent manner, suggesting that the mutant Rad54 protein binds and stays at recombination sites. The *rad54-K300A* mutation increases isochromosome formation to the same extent as the *rad54* deletion [117], showing that Rad54 ATP-dependent activity is vital in suppressing isochromosome formation. After DNA strand invasion, Rad54 ATPase dissociates Rad51 from D-loops and allows DNA polymerase to bind the 3′-OH end of the invading strand to initiate DNA repair synthesis [149]. ATP-dependent Rad54 motor facilitates Rad51-mediated D-loop formation but can also dissociate D-loops [150,151]. These reactions mediated by Rad51 and Rad54 are essential steps of synthesis-dependent strand annealing (SDSA) that result only in noncrossover recombination. In contrast to other types of homologous recombination, noncrossover recombination does not result in changes in chromosomal structure. Homologous recombination mediated by Rad51 and Rad54 preferentially promotes noncrossover and maintains genome integrity. 

In the meiotic prophase, crossover recombination is induced to ensure faithful segregation of homologous chromosomes and to produce genetic diversity. In this case, a meiosis-specific Rad51 paralog, Dmc1, rather than Rad51, plays an essential role in joint molecule formation [152]. Rather than Rad54, another SWI2/SNF2-type motor protein, Tid1/Rdh54, facilitates Dmc1-mediated DNA joint molecule formation [153]. Meiosis-specific recombination machinery promotes crossover recombination between homologous chromosomes.

### 5.3. Centromere-Specific Control of DNA Recombination: Rad51-Dependent Recombination and Noncrossover

Distinct pathways of homology-mediated DNA recombination exist. Yeast Rad52 facilitates Rad51 nucleoprotein filament formation onto single-stranded DNA decorated with single-strand DNA-binding protein, RPA [154,155,156]. Rad52 also has the Rad51-independent activity to anneal single-stranded DNA strands of complementary sequences, the so-called single-strand annealing (SSA) [157]. In mammals, BRCA2, but not Rad52, plays a major role in forming Rad51 nucleoprotein filaments. However, like yeast Rad52, mammalian Rad52 has the SSA activity. Rad52 also has homologous DNA pairing activity to produce D-loops [158,159]. Furthermore, Rad52 facilitates the annealing of DNA and RNA strands to form DNA-RNA hybrids and the inverse strand exchange responsible for DSB repair using an RNA strand as the template [160]. Rad52 is a critical player in Rad51-independent recombination. Hereafter, we refer to Rad51-independent but Rad52-dependent recombination as Rad52-dependent recombination or SSA for simplicity. 

Importantly, the choice of recombination pathways at a centromere differs from that in non-centromeric regions. Recombination between inverted DNA repeats can occur in either the Rad51- or Rad52-dependent pathway in arm regions. However, at a centromere, the inverted repeat recombination occurs exclusively in the Rad51-dependent pathway in the fission yeast [60]. The factors involved in DNA replication elongation suppress Rad52-dependent recombination at centromeres [122], suggesting that restricted lengths of single-stranded DNA formed during centromere replication prevent Rad52-dependent SSA. However, what induces centromere-specific recombination pathway selection remains unknown.

Noncrossover recombination is predominant at a centromere compared to an arm region. The Rad51-dependent pathway preferentially promotes noncrossover recombination. However, it is possible that Mhf1 and Mhf2 further enhance noncrossover recombination at centromeres. Mhf1 and Mhf2, also known as CENP-S and CENP-X, contain the histone-fold domain and play roles in DNA repair and chromosome segregation (Table 1 and Table 2). Mhf1 and Mhf2 interact to form Mhf1-Mhf2 heterotetramers and bind DNA joint molecules [161]. The Mhf1-Mhf2 complex recruits the Fanconi anemia complementation group M (FANCM) DNA helicase to the joint molecule, leading to the SDSA pathway that results only in noncrossover recombination [59,162,163,164]. Mutations in Mhf1, Mhf2, or Fml1/FANCM helicase greatly increased crossover recombination at a centromere and isochromosome formation in fission yeast [60]. Remarkably, Mhf1 and Mfh2 localize at centromeres and form a nucleosome-like structure with CENP-T and CENP-W, ensuring faithful chromosome segregation [130,131,132,165]. The centromere localization of Mhf1 and Mhf2 may also facilitate noncrossover recombination at the centromere.

### 5.4. The Rad51 Function at Centromeres throughout the Cell Cycle

Rad51-dependent homologous recombination is thought to occur only in the S and G2 phases of the cell cycle. However, a recent study demonstrated in mammalian cells that Rad51 works in the G1 phase at a centromere to repair DSBs despite the absence of sister chromatids [125]. Centromeres contain active chromatin marks, including di-methylation of histone H3 4th lysine (H3K4me2) [85,166]. In response to DSBs, H3K4me2 induces centromeric transcription and DNA-RNA hybrid formation, facilitating DNA-end resection [167,168,169]. CENP-A, HJURP, and Mis18 are involved in recruiting Rad51 to the centromere upon DSB formation in G1 cells [125]. CENP-A and HJRUP interact with the USP11 deubiquitinase. USP11 removes ubiquitins from PALB2 to promote its interaction with BRCA1 and BRCA2 and recruit Rad51 to DSB sites [170]. HJURP has been identified as the protein involved in the homologous recombination pathway of DSB repair through interaction with NBS1, a component of the MRN complex [62]. The HJURP protein directly binds Holliday junctions and is overexpressed in many cancers [62,171,172,173,174,175,176]. The DNA-binding activity of HJURP might affect centromere function and integrity. With the aid of the centromere-specific proteins, Rad51 plays a role in maintaining centromere integrity even in the G1 phase.

Rad51 plays a role at a centromere also in the M phase. A centromere is a late-replicating region of the genome and consists of many repetitive sequences, making it susceptible to a high level of recombination in human cells [57,86]. Mitotic DNA synthesis (MiDAS) occurs at a centromere to repair DNA damage persisting in the M phase. Phosphorylation of Rad51 at S14 by the key mitotic kinase Polo-like kinase (PLK) facilitates Rad51 to maintain centromere integrity during MiDAS [177]. Rad51 has access to the centromere throughout the cell cycle.

### 5.5. A Role of Rad51-Dependent Homologous Recombination in Centromere Chromatin

In fission yeast, *rad51* deletion increases levels of chromosome loss and causes hypersensitivity to a microtubule destabilizing drug, thiabendazole [105]. Rad51 and Rad54 are partially required for transcriptional gene silencing at a centromere [117], suggesting a role of Rad51-dependent recombination in establishing or maintaining centromere heterochromatin. It has also been suggested that the Rad51-dependent recombination between centromere repeats forms high-order chromatin structures in favor of centromere function [178]. In *C. albicans*, Rad51 and Rad52 coimmunoprecipitate with CENP-A [179]. Chromatin immunoprecipitation (ChIP) showed that CENP-A binding levels at centromeres are reduced in the absence of Rad51 or Rad52 [179], suggesting that the Rad51-dependent recombination pathway stabilizes the CENP-A chromatin. In human cells, DNA strand breaks accumulate at active αSat HORs in replication-dependent and -independent manners [126]. The type II topoisomerase TOP2B and the type I topoisomerase TOP3A are involved in centromeric DNA break formation. Rad51 is required to repair the centromere breaks and for stable localization of CENP-A to the centromere [126,177]. These lines of evidence suggest a role of the Rad51-dependent recombination in maintaining or formation of the centromere chromatin. However, the detailed mechanism by which Rad51 contributes to the centromere chromatin remains unknown.

## 6. The Mechanism of GCR at Centromeres

### 6.1. Single-Strand Annealing (SSA) Causes GCR at Centromeres

Centromeric GCR occurs in a manner independent of Rad51. In fission yeast, Rad52 causes isochromosome formation through a single-strand annealing (SSA) [122] (Figure 5). Amino acid substitution of the conserved residue in the DNA-binding domain of Rad52, *rad52-R45K*, reduced Rad52 SSA activity in vitro. Unlike *rad51* or *rad52* deletion, the *rad52-R45K* mutation does not increase GCR rates in the otherwise wild-type background, showing that the *rad52-R45K* mutation does not impair the Rad51 loading function of Rad52. However, in the *rad51* deletion background, the *rad52-R45K* mutation greatly reduces isochromosomes but not chromosomal truncates, suggesting that Rad52 SSA specifically causes centromeric GCR, resulting in isochromosome formation. Rad52 SSA may use single-stranded DNA produced during DNA replication as an annealing partner. Mutations in DNA replication elongation factors, including DNA Pol α, Polε, Swi1/Tof1/Timeless, Pof3/Dia2/STIP1, but not the initiation factor such as Cdc18/Cdc6, induce Rad52-dependent centromere recombination and isochromosome formation [122]. In DNA mismatch repair (MMR), Msh2–Msh3 and Msh2–Msh6 heterodimers recognize DNA loops and mismatches, respectively, and recruit Mlh1. In addition to its role in MMR, Msh2–Msh3 binds joint molecules and facilitates SSA [180,181]. Msh2 and Msh3, but not Msh6 and Mlh1, were found to promote Rad52-dependent isochromosome formation [122]. Msh2-Msh3 heterodimers may stabilize the joint molecule formed by Rad52 SSA.

Ubiquitination of proliferating cell nuclear antigen (PCNA) plays a role in Rad52-dependent isochromosome formation. PCNA homotrimers form a DNA-sliding clump acting as a landing pad for DNA replication, recombination, and repair factors [182,183]. Post-translational modification of PCNA is critical in DNA transactions [184]. Mono-ubiquitination of PCNA at the 164th lysine (K164) recruits the DNA polymerases involved in translesion synthesis. On the other hand, K164 poly-ubiquitination promotes template switching, a recombination-mediated damage bypass pathway [185]. In the budding yeast *cdc9* mutant of DNA ligase I, PCNA K107 is ubiquitinated in a manner dependent on Rad5 ubiquitin ligase and Mms2-Ubc4 ubiquitin-conjugating E2 enzymes [186]. The PCNA K107R mutation is lethal in the *cdc9* mutant, suggesting a role for PCNA K107 ubiquitination when unligated Okazaki fragments are accumulated. Fission yeast Rad8 ubiquitin ligase is the homolog of budding yeast Rad5 and mammalian HLTF. Rad8 and its homologs contain HIRAN, RING-finger, and SWI2/SNF2 translocase domains. HIRAN binds DNA 3′-end [187,188,189], while RING-finger interacts with ubiquitin-conjugating E2 enzymes [190,191]. Rad8 promotes isochromosome formation in the absence of Rad51. Mutations in HIRAN or RING-finger but not translocase domains reduced isochromosome formation [121]. Mutations in Mms2 or Ubc4 but not Ubc13 E2 enzymes also reduced isochromosome formation [121]. PCNA K107R but not K164R mutation reduced isochromosome formation [121,186]. It appears that Rad8 binds the 3′-end of single-stranded DNA and recruits the Mms2-Ubc4 E2 enzyme to the recombination site to ubiquitinate PCNA at K107. PCNA K107R and *rad52-R45K* mutations epistatically reduce isochromosome formation, showing that PCNA K107 and Rad52 act in the same GCR pathway. However, it is unknown how the PCNA K107 ubiquitination facilitates Rad52-dependent GCR. As K107 is located at the PCNA-PCNA interface, K107 ubiquitination may change the PCNA ring structure to stabilize joint molecules or recruit the factor(s) involved in Rad52-dependent GCR.

Recently, it has been shown that Skb1/PRMT5 arginine methyltransferase (RMTase) and Srr1 play roles in isochromosome formation in the absence of Rad51 [136]. Mutation in the RMTase domain of Skb1 reduces GCR rates in *rad51* deletion cells [136], showing that Skb1 causes isochromosome formation through RMTase activity. Skb1 regulates cell morphology and cell cycle progression, respectively, with Slf1 and Pom1 [192,193,194,195]. However, neither Slf1 nor Pom1 was required for isochromosome formation, indicating that Skb1 causes isochromosome formation through the function independent of the cell morphology and cell cycle control. Skb1 acts in the Rad52-dependent GCR pathway, as *skb1* deletion does not further reduce GCR rates in *rad52-R45K* mutant cells. Finding Skb1 RMTase targets is vital to elucidate how Skb1 causes isochromosome formation. Remarkably, *skb1* and *srr1* deletions additively reduce GCR rates [136], showing that Skb1 and Srr1 have nonoverlapping roles in GCR. In contrast to *skb1*, a *srr1* mutation further reduces GCR rates in *rad52-R45K* mutant cells, suggesting a role of Srr1 in the Rad52-independent GCR pathway. *srr1* and *rad51* mutations synergistically increase sensitivity to DNA-damaging agents, including methyl methanesulfonate, camptothecin, and hydroxyurea [136], suggesting a role of Srr1 in Rad51-independent DNA damage repair. However, how Srr1 acts in DNA damage tolerance and causes isochromosome formation remains unknown.

### 6.2. Crossover and Break-Induced Replication (BIR) Cause GCR at Centromeres

Crossover recombination and break-induced replication (BIR) can result in isochromosome formation (Figure 5). In the absence of Rad51, spontaneous isochromosomes are produced mainly through crossover recombination but not BIR. Mus81 endonuclease preferentially promotes crossover resolution of joint molecules [146,196,197,198], whereas DNA polymerase δ (Polδ) promotes BIR [199,200]. In fission yeast *rad51* deletion cells, Mus81 is required for the spontaneous formation of isochromosomes [117,122]. However, Cdc27/Pol32/PolD3, the third largest subunit of Polδ, is not essential for the isochromosome formation [117]. 

DSBs introduced outside a centromere region induce isochromosome formation, following an extensive degradation of DSB ends reaching a centromere [106,113]. In this case, Cdc27 is required for the isochromosome formation, suggesting that BIR produces DSB-induced isochromosomes. The reason for the discrepancy remains unknown. However, the nature of the initial DNA damage (i.e., spontaneous damage or extensively end-resected DNA) may affect later steps of isochromosome formation.

At common fragile sites (CFSs) of mammalian chromosomes, Rad52 causes MiDAS [99,201,202,203]. Interestingly, Rad52 recruits Mus81-Eme1 and PolD3 to the CFS loci, suggesting that Rad52 induces both crossover and BIR.

### 6.3. Microhomology-Mediated End-Joining (MMEJ) in Centromeric GCR

Microhomology-mediated end-joining (MMEJ) is also involved in centromeric GCR. DNA polymerase θ (Polθ), a member of the A family, plays a critical role in MMEJ [204,205,206,207]. In mammals, CRISPR-Cas9-induced DSBs at centromeres recruit Rad51 to the centromere even in the G1 phase [125]. Inhibition of Rad51 increases the number of chromosomes with broken centromeres and translocation. In the absence of Rad51, Rad52 localizes at centromeric DSBs and facilitates the GCR, suggesting a role of Rad52 in centromere-repeat-mediated GCR. In addition to Rad52, Polθ is required for centromeric translocation [125]. SSA and MMEJ are the pathways that cause centromeric GCR in mammalian cells. Although no Polθ homologs in yeast, centromeric GCR may also occur through MMEJ or a similar mechanism [208,209,210], as *rad52* mutation does not eliminate isochromosomes.

## 7. Perspectives

In the past few decades, it has been shown that the centromere region of chromosomes containing repetitive DNA sequences is one of the fragile sites in the eukaryote genome. The CENP-A chromatin provides the unique feature of the centromere. The CENP-A chromatin works as the landing pad for the kinetochore assembly and as a critical regulator of DBS repair. Pericentromeric heterochromatin is another crucial aspect of the centromere. However, the detailed mechanism by which these epigenetic chromatin structures affect centromere integrity is unknown. It is generally assumed that heterochromatin represses DNA transactions. However, heterochromatin is a dynamic genomic environment, and DNA recombination and transcription can occur in heterochromatin [105,211,212,213]. It is of great interest to understand how DNA recombination, repair, replication, and transcription at the centromere are controlled to maintain centromere and genome integrity and how DNA transactions affect, in turn, the centromere function in chromosome segregation.

## Figures and Tables

**Figure 1 biomolecules-14-00028-f001:**
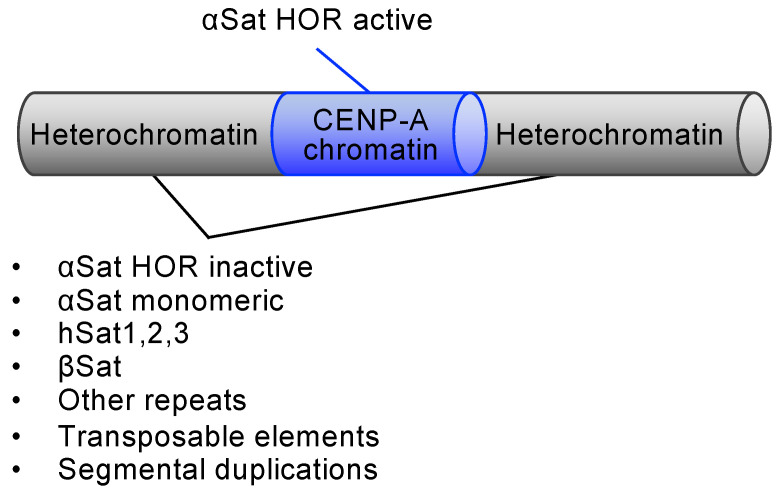
A schematic overview of chromatin structures and DNA sequences of human centromeres.

**Figure 2 biomolecules-14-00028-f002:**
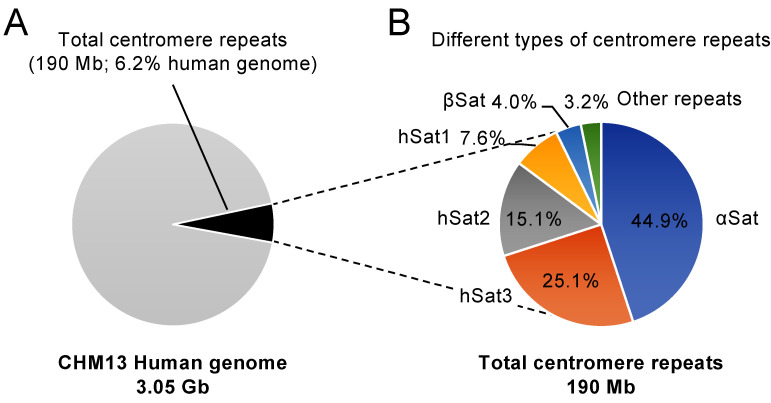
Human centromere repeats. (**A**) The percentage of total centromere repeats in the Telomere-to-Telomere (T2T) CHM13 human genome [13]. A complete hydatidiform mole (CHM) arises from the loss of the maternal complement and duplication of the paternal complement postfertilization. Therefore, most CHM genomes are homozygous diploids. CHM13 is a cell line originally isolated from a CHM at Magee-Women’s Hospital (Pittsburgh, PA, USA). (**B**) The percentage of each type of satellite repeat sequence [22] in the total centromere repeats of 190 Mb.

**Figure 4 biomolecules-14-00028-f004:**
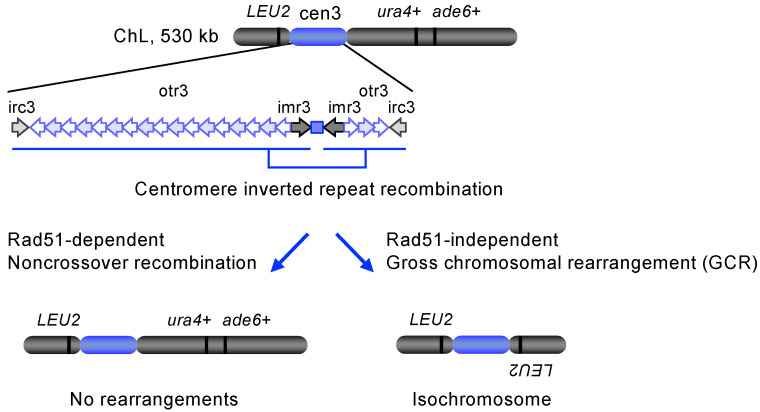
Isochromosome formation in fission yeast. Isochromosomes that are otherwise lethal in haploid cells can be detected using an extra-chromosome ChL derived from chr3 [105,116]. Positions of *LEU2*, *ura4*+, and *ade6*+ genetic makers that are used to monitor the ChL integrity are indicated. Rad51 promotes conservative noncrossover recombination between centromere repeats (the left arrow). On the other hand, Rad51-independent GCR results in isochromosome formation (the right arrow).

**Figure 5 biomolecules-14-00028-f005:**
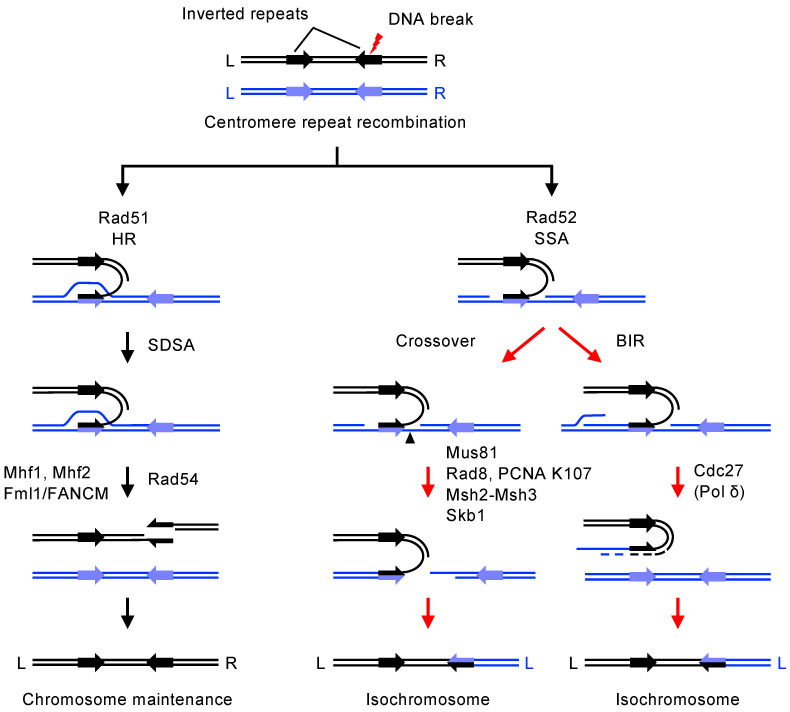
Centromere recombination pathways that maintain chromosome integrity or cause isochromosome formation in fission yeast. HR, homologous recombination; SDSA, synthesis-dependent strand annealing; SSA, single-strand annealing; BIR, break-induced replication.

**Table 2 biomolecules-14-00028-t002:** DNA repair and recombination proteins that have roles at centromeres.

Mammals	Fission Yeast	DNA Transaction	Centromere
RAD51	Rad51	HR ^2^	Localization and GCR suppression [105,117,125,126]
BRCA1	– ^1^	HR	Localization and GCR suppression [125,127]
BRCA2	–	HR	GCR suppression [125,128]
PALB2	–	HR	GCR suppression [125]
RAD51C	Rad55	HR	GCR suppression [113]
XRCC3	Rad57	HR	GCR suppression, Chromosome segregation [113,129]
RAD54	Rad54	HR	GCR suppression [117]
XRCC2	Rlp1	HR	Chromosome segregation [129]
FANCM	Fml1	HR	GCR suppression [60]
CENP-S	Mhf1	HR	Localization and GCR suppression [60,130,131,132]
CENP-X	Mhf2	HR	Localization and GCR suppression [60,130,131,132]
MRE11	Mre11	HR, NHEJ ^3^	Localization and GCR suppression [113,133]
RAD50	Rad50	HR, NHEJ	Localization and GCR suppression [113,133]
NBS	Nbs1	HR, NHEJ	Localization and GCR suppression [113,133]
ATR	Rad3	DNA damage checkpoint	Localization and GCR suppression [105,107]
ATRIP	Rad26	DNA damage checkpoint	Localization and GCR suppression [107]
DNA2	Dna2	HR	Localization and centromeric replication [134]
ERCC6L2	–	NHEJ	Localization and centromeric replication [135]
RAD52	Rad52	HR, SSA ^4^, DNA pairing	GCR suppression and GCR [60,117,122,125]
MUS81	Mus81	HR	GCR [122,60]
PCNA	Pcn1	HR, Replication, Repair	GCR [121]
HLTF	Rad8	HR	GCR [121]
EXO1	Exo1	HR	GCR suppression [113]
BLM	Rqh1	HR	GCR [113]
POLD3	Cdc27	Replication, HR	GCR [113]
PRMT5	Skb1	HR, NHEJ	GCR [136]
SRRD	Srr1	DNA damage tolerance	GCR [136]
Polθ	–	MMEJ ^5^	GCR [125]
MSH2	Msh2	MMR ^6^, HR	Localization and GCR [122,137]
MSH3	Msh3	MMR, HR	Localization and GCR [122,137]

^1^ No homolog in fission yeast. ^2^ HR, Homologous recombination. ^3^ NHEJ, Non-homologous end-joining. ^4^ SSA, Single-strand annealing. ^5^ MMEJ, Microhomology-mediated end-joining. ^6^ MMR, DNA Mismatch repair.

## Data Availability

Not applicable.

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
