# Peer review of "Gross Chromosomal Rearrangement at Centromeres"

_biomolecules, 2023, doi:10.3390/biom14010028_

Round 1

Reviewer 1 Report

Comments and Suggestions for Authors

The review manuscript delves on the centromere integrity and the mechanisms behind centromere rearrangements, providing a comprehensive overview of the current literature on this subject in both humans and fission yeast. 

Overall the manuscript is well written effectively capturing the current state of the art.

However, in my opinion the manuscript should be improved by providing discussion and citation of very recent papers, which can be already considered as important milestones in the field. 

The introduction should be expanded. I think that the massive presence of transposable elements and satellite DNA in a variety of eukaryotic organisms should be highlighted in the introduction.  A couple of review are worth of citation in this context (10.3390/cells11050761; 10.3390/cells9122714). Moreover, the recent important advancement (T2T) in genome sequencing and assembly (including centromeric and pericentromeric regions) should be mentioned in the introduction (10.1126/science.abj6987; 10.1038/s41586-023-06457-y; 10.1016/j.yexcr.2020.112127).

Finally, authors should also discuss a recent paper analyzed the recombination signals between heterologous human acrocentric chromosomes related to the occurrence of Robertsonian translocation (10.1038/s41586-023-05976-y). 

Other issues.

l37-38. the DNA sequence at the centromere is not conserved at all.

figure 2 is not easy to understand. What do the percentage in the middle of the pictures refer to?

figure 2 caption. what is CHM13?

Figure 4. This picture is not easy to understand. What do bars represent? Please, provide detailed description of all the elements in you figures.

l259 Remove "whose arms are mirror images of each other" (the definition of isochromosome has been given earlier in the text)

l264-266 This paragraph is not related to fission yeast. Move it to a more appropriate section and add references.

l81 "(Figure. 2)" remove dot

Reviewer 2 Report

Comments and Suggestions for Authors

In the manuscript “Gross Chromosomal Rearrangement at Centromeres” by Xu et al., the authors reviewed different measures to suppress chromosomal rearrangements at centromeres, including homologous recombination, single-strand annealing, and microhomology-mediated end-joining. The introduction of the chromatin structures and DNA repeats at centromeres provides an excellent description of the centromeres with sufficient details of the chromatin and heterochromatin regions. The roles of Rad51, Rad52, and Rad54 in chromosomal rearrangement at centromeres have been clearly summarized for the general audience to understand and follow.

Here are the detailed comments and suggestions:

1.     Fig. 2 needs to be clarified for the zoomed-in pie chart part. On the right side, the percentage does not match the color-coded types of satellite repeat sequences. It would be nice for the authors to change the black pie piece on the left for centromere repeats to a colored version to represent the different types of repeats.

2.     Including a sub-panel in Fig. 3 to illustrate the difference between dg and dh would be great.

3.     Fig. 4 does not clearly illustrate the difference between chromosome maintenance and isochromosome. The authors refer the readers to the figure in the text on page 8, line 272, but the figure needs to be clarified.

4.     Page 9, line 320, “Rad54 ATP-dependent branch migration can dissociate D-loops, an essential step of synthesis-dependent strand annealing (SDSA) that results only in noncrossover recombination.” Wright & Heyer Mol. Cell2014 has demonstrated that Rad54 acts as a heteroduplex pump to drive D-loop formation and clear Rad51 for new repair DNA synthesis.

           5. Page 9, line 323, “Intriguingly, during the meiotic prophase where crossover recombination is induced, 323 a meiosis-specific Rad51 paralog, Dmc1, rather than Rad51, plays an essential role in joint 324 molecule formation [142].” Dmc1 is the RecA or Rad51 homolog, which catalyzes DNA homology search and strand invasion in meiosis. I don’t understand why it is intriguing. 

Comments on the Quality of English Language

Some editing would be great. For example, page 7 line 254, "breast and ovarian cancer" should be "breast and ovarian cancers."

Round 2

Reviewer 1 Report

Comments and Suggestions for Authors

I thank the Authors for their effort in addressing my comments. 

I have just one final comment. The discussion could benefit from a short paragraph elucidating the contrast between the low frequency of recombination at centromeric DNA (a well-known characteristic of heterochromatic loci) and the evidence supporting the role of recombination in generating GCRs in centromeres. Could the absence of recombination be considered an additional misconception associated with heterochromatin? There is indeed evidence that heterochromatin is a dynamic genomic environment often linked to transcription and gene expression(reviewed in 10.1007/s00412-006-0052-x; 10.1111/febs.15104; 10.1016/j.tig.2019.06.002)

l354. Meiosis-specific recombination machinery promotes crossover recombination between homologous chromosomes" this sentence is not dispensable since a description has been given in the paragraph above.

Comments on the Quality of English Language

Minor editing of English language required
